# FGFR3-TACCs3 Fusions and Their Clinical Relevance in Human Glioblastoma

**DOI:** 10.3390/ijms23158675

**Published:** 2022-08-04

**Authors:** Hanna Gött, Eberhard Uhl

**Affiliations:** Department of Neurosurgery, University Hospital Giessen, 35398 Giessen, Germany

**Keywords:** FGFR3-TACC3, oncogenic fusion, glioblastoma, tyrosine kinase receptors, FGFR, TACC, molecular signaling, molecular diagnostics

## Abstract

Oncogenic fusion genes have emerged as successful targets in several malignancies, such as chronic myeloid leukemia and lung cancer. Fusion of the fibroblast growth receptor 3 and the transforming acidic coiled coil containing protein—FGFR3-TACC3 fusion—is prevalent in 3–4% of human glioblastoma. The fusion protein leads to the constitutively activated kinase signaling of FGFR3 and thereby promotes cell proliferation and tumor progression. The subgroup of FGFR3-TACC3 fusion-positive glioblastomas presents with recurrent clinical and histomolecular characteristics, defining a distinctive subtype of IDH-wildtype glioblastoma. This review aims to provide an overview of the available literature on FGFR3-TACC3 fusions in glioblastoma and possible implications for actual clinical practice.

## 1. Introduction

In the last decades, the clinical field of oncology has been impacted by advances in molecular diagnostics and specifically targeted therapy approaches that not only changed our perception of oncology practice, but also offered hope to patients, whose fate would have previously been fatal. The most prominent example for the successful story of personalized medicine is the discovery of the BCR-ABL1 fusion gene that is formed via interchromosomal rearrangement in 95% of patients with chronic myeloid leukemia (CML). The formation of BCR-ABL1 fusion is possible through a chromosomal translocation of genetic material from chromosome 22 and chromosome 9, building the famous Philadelphia chromosome [1]. The BCR-ABL1 fusion gene codes for a constitutively activated tyrosine kinase and can be inhibited by treatment with specific inhibitors, such as imatinib, which was able to improve the prognosis of CML from a mean overall survival of 3–7 years to a 10 year mean survival rate of more than 80% [2]. Similarly, the EML4-ALK fusion, which is prevalent in 3–5% of patients with non-small cell lung cancer (NSCLC), is a target for tyrosine kinase inhibitors, and the fusion targeting therapy improved the prognosis of these patients significantly [3,4]. Currently, a number of other oncogenic fusion genes have been described in various malignancies.

Contrary to the trend of personalized medicine in cancer treatment and immense research effort and molecular characterization, there has not yet been a successful trial for targeted therapy in glioblastoma. The treatment of malignant glioma is based on gross total resection, if possible, followed by concomitant chemoradiation and adjuvant chemotherapy with temozolomide [5,6]. The mean overall survival rate in glioblastoma patients ranges between 14 to 16 months with maximal treatment [7,8,9]. Most studies that showed a survival benefit in glioblastoma patients focused on the treatment with systemic cytotoxic chemotherapy protocols or other systemic approaches, such as the application of tumor treating fields [8,9].

In 2012 Singh et al. were the first to discover a gene fusion of the fibroblast growth factor receptor 3 (FGFR3) and the transforming acidic coiled coil containing protein 3 (TACC3) in human glioblastoma [10]. Since then, several studies have found the FGFR3-TACC3 fusion gene as a susceptible target in a subset of other cancer types. Furthermore, an increasing number of FGFR-inhibitors have made their way into clinical trials. This increases the hope that FGFR3-TACC3 fusion in human glioma might serve as a potential target for personalized diagnostics and specific therapy.

With this review, we aim to provide an overview of the pathophysiology of oncogenic FGFR-TACC fusion and a current update on the diagnostic and clinical relevance of the FGFR3-TACC3 fusion in human glioma.

## 2. Glioblastoma

Glioma are the most frequent malignant primary CNS tumors in adults, with an incidence of 5–6 per 100,000 per year, with glioblastoma (with 3.2 per 100,000 per year) being the largest subgroup [11]. Males are 1.6-fold more often affected than females, and a higher incidence in Caucasians with higher socioeconomic status is reported [11], although this might be biased by social inequality in access to medical care. Today, no risk factors except prior exposure to radiation are known [12]. The mean at diagnosis is 64 years of age, and the highest incidence is in the group between 74–85 years of age [11].

After an MRI scan, the diagnosis is confirmed by stereotactic biopsy or tumor resection, if feasible [6]. With the current updated WHO-classification of CNS tumors from 2016 and the subsequent recommendations from cIMPACT-NOW, an integrated histomolecular diagnosis is obtained. Besides the histomorphology of an infiltrating diffuse astrocytic tumor, several biomarkers are required to confirm the diagnosis of a glioblastoma: astrocytic tumors, presenting with necrosis or microvascular neoproliferation in the absence of an IDH-mutation and wildtype histone H3 status are characterized as IDH-wildtype glioblastoma, WHO grade 4 [13,14]. IDH-mutant tumors, presenting with the histology of a WHO grade 4 tumor, are no longer referred to as glioblastoma, but as astrocytoma, IDH-mutant, WHO grade 4 instead [13,14]. Common genetic alterations that are associated with glioblastoma are EGFR amplifications, mutations in the promotor region of TERT, and a gain on chromosome 7, in addition to a loss on chromosome 10 [15]. An important predictive marker for the response towards treatment with alkylating chemotherapy is the MGMT promotor methylation, which is present in about 45% of glioblastoma and can help in clinical decision making [16].

The treatment of glioblastoma is based on maximal save resection of the tumor, followed by concomitant radiochemotherapy with temozolomide and additional adjuvant chemotherapy with six cycles of temozolomide, according to the STUPP-regimen [5]. In Germany, the alternative CeTeG-regimen, with a combination of CCNU and temozolomide, is used for the treatment of young patients with MGMT-promotor methylated glioblastoma [7]. In recent years, the therapeutic options for glioblastoma have been expanded by the implementation of tumor treating fields into clinical practice [9]. Nevertheless treatment options for glioblastoma are still limited, and the five-year survival rate is only 5.5% [17].

Despite an immense research effort, today, no targeted therapeutic approach for the treatment of glioblastoma is available. The VFGR antibody bevacizumab did not increase the overall survival rate for glioblastoma, and its use is limited to transient treatment for symptom control [6,18]. The mean overall survival in glioblastoma patients ranges from 14 to 16 months [9,17].

## 3. The FGFR Family and FGFR Signaling

The family of fibroblast growth factor receptors (FGFRs) consists of four membrane-bound tyrosine kinase receptors (RTK) FGFR1–4 and one kinase-lacking coreceptor FGFR5 or FGFR-like1 (FGFRL1). The structure among the FGFR1–4 family members is highly conserved: every FGFR consists of three immunoglobulin-like looped domains (Ig-domains), an acidic region (the acid box), a single transmembrane domain, a long juxtamembrane domain, and an intracellular domain with two tyrosine kinase domains [19]. Alternative splicing of the third Ig domain of FGFR1–3 creates two distinct isoforms of the receptors, named FGFR(1–3)IIIb and IIIc, with different ligand affinities and tissue specificity [20,21]. While FGFR(1–3)IIIb is usually expressed in epithelial tissue, FGFR(1–3)IIIc is more prevalent in a mesenchymal environment [22]. Although splicing regulatory mechanisms are not yet fully understood, the switching of the expression between the isoforms during oncogenic or developmental processes, such as epithelial to mesenchymal transition, is possible [23].

Via the extracellular membrane domain, FGFRs can bind 22 known fibroblast growth factor ligands (FGF), as well as the FGFR-coreceptors heparan sulfate proteoglycans and klotho. In addition to these canonical binding partners, FGFRs are also able to bind not only extracellular matrix associated proteins and cell adhesion molecules, but also a broad spectrum of non-canonical signaling partners such as N-cadherins or galectins [24,25]. The numerous possible ligand-receptor combinations point out that FGFR-signaling is highly context-specific, and the role of the same receptor in two distinct tissues or pathologies can be completely different.

Ligand binding at the extracellular IgII and IgIII binding domain induces the homodimerization of two FGFR molecules, and interaction between their tyrosine kinase domain leads to the transautophosphorylation and initiation of downstream signaling pathways (Figure 1) [20,26,27,28]. The most relevant FGFR-downstream pathways in cancer are activated by the FGFR substrate 2 (FRS2) mediated binding of growth factor receptor bound protein 2 (GRB2). Binding of the RAS guanine exchange factor, son of sevenless homologue 1 (SOS) and the cofactor GRB2-associated-binding protein 1 (GAB1), forms a complex that activates RAS/MEK/MAPK and RAC/JNK and PI3K/AKT signaling. All these pathways result in activation of cell proliferation, cell survival, and STAT1 and 3 activation, resulting in the induction of gene translation [29,30].

In addition to other mechanisms, FGFR-signaling is negatively regulated by miRNA-mediated degradation of the receptor [29,31] and SPROUTY (SPRY) proteins, which function as tumor suppressor genes by negative regulation of RAS/MAPK-activity [26,29,32,33].

The central function of FGF-FGFR in pathways related to cell proliferation and survival in combination with the variable and tissue specific complex regulatory loops implicate the consequences of disturbance of these mechanisms in cancer.

A whole genome analysis of more than 2662 adult human glioblastoma samples revealed the number of FGFR-aberrations and amplifications as generally sparse, with prevalence of FGFR1 aberrations of 51 in 3068, FGFR2 in 12 of 2662, FGFR3 in 13 of 2887, and FGFR4 in 9 out of 2456 investigated samples [34]. While little is known about the role of FGFR2 and FGFR4 in glioblastoma, FGFR1 is expressed in human glioma, and its expression level increases with the grade of malignancy [35]. The expression of stem cell transcription factors such as ZEB1, SOX2, and OLIG3 is regulated by FGFR1 [36]. FGFR3 expression is increased in the classical and neural subtype of glioblastoma, and Wang et al. saw improved survival and an expressome associated with differentiated cellular function in correlation with higher FGFR3 expression [37].

## 4. FGFR-Fusions in Cancer

Fusion proteins are hybrids of two formally separate genes, encoding for two separate proteins. They result from interchromosomal—as in case of the Philadelphia Chromosome—or intrachromosomal gene rearrangements. After the initial detection of a FGFR3-TACC3 fusion in a small subset of human glioblastoma [10], the search for FGFR alterations, and especially FGFR-fusion proteins, in cancer has been a subject of significant research.

FGFR-alterations are common in various types of cancer, with a frequency of 7–9.2% in more than 47 different histological types among a large series [27,38,39]. Th most affected genes are FGFR1-4, with amplifications being the most common, followed by mutations [38,39]. FGFR-fusions account for about 10% of all FGFR-alterations in cancer [39]. Wu et al. identified 24 cases with FGFR1-3 fusions in RNA sequencing data from 322 tumor samples affecting cholangiocarcinoma, breast, prostate, and thyroid cancer, as well as lung squamous cell carcinoma, bladder cancer, head and neck cancer, and glioblastoma [40]. These results were confirmed by the work of others, who found FGFR-fusion genes in the tumor types mentioned above, as well as in gastric and colorectal carcinoma [22,38,41,42,43,44,45,46]. There are two types of FGFR-fusions observed. In the first type, which is the more frequent version, 3′-FGFR is fused with a 5′-fusion partner as, for example, BICC1, AFF3, CASP7, CCDC6, KIAA1967, OFD1, BAIAP2L1, or TACC3. In the second type, 3′-partners, such as SLC45A3, BAG4, or ERLIN2 fused to 5′ FGFR, are described [38,39,40,47].

FGFR1-fusions are relatively rare. However, FGFR1-TACC1 fusions can be detected in GIST, glioblastoma, lower grade glioma, and other tumors of the central nervous system [38,48,49,50,51]. A FGFR1-NTM fusion can be found in bladder cancer [38,48], and fusion was described in NSCLC a BAG4-FGFR1 [43].

FGFR2-fusions are the most frequent FGFR-fusions in solid cancer and can be found in cholangiocarcinoma, breast cancer, prostate cancer, thyroid cancer, lung cancer, bladder cancer, oral and head and neck cancer, as well as in glioblastoma [39,40,52,53,54,55]. They are associated with a better overall survival in cholangiocarcinoma [56,57].

Although there are more than 100 different fusion partners, with the greatest variety among FGFR2-fusions, most of the fusion partners are known for their ability of dimerization or oligomerization [40,58]. Furthermore, all fusion proteins had intact kinase domains, and autophosphorylation of the FGFR-fusion proteins by oligomerization, leading to constitutive upregulation of MAPK/ERK and PI3K signaling, is a commonly observed phenomenon [20,29,40,49].

Fusion of the fibroblast growth receptor 3 (FGFR3) gene to the transforming acidic coiled coil containing the protein 3 (TACC3) gene leads to formation of the FGFR3-TACC3 fusion gene. The hybrid gene codes for one of the most recurrent oncogenic fusion proteins and can be found in about 3% of squamous NSCLC and urothelial carcinoma, 3.9% of cervical cancer patients, and in 4% of glioblastoma patients [39,42,59,60]. A FGFR3-BAIAP2L1 fusion was also found in bladder and lung cancer [22,61].

The FGFR3 gene and the TACC3 gene are located on Chromosome 4p16.3, only 48kb apart from each other within a region that is associated with translocation causing breakpoints in multiple myeloma [62,63]. This spatial proximity and the localization at a double-strand hotspot is one theory explaining why this fusion protein is found so frequently. An alternative explanation of the frequent recurrence of the FGFR3-TACC3 fusion is that the fusion protein leads to specific survival benefits, such as proliferation gain and increased cell survival. Cells that express the fusion protein might thereby prevail in a setting of selection pressure [20].

There is high variation among the breaking points of the FGFR3-TACC3 fusion protein, but all variants share the same functional architecture: The 3′ partner of the fusion, the FGFR3-gene, is fused with the 5′-partner, the TACC3 gene, via tandem duplication (Figure 2) [64]. The kinase domain of FGFR3 remains intact in all observed FGFR3-TACC3 fusion variants. One of the key functions of coiled coil domains is protein–protein interaction and thereby, mediating oligomerization [65,66]. While the FGFR3-wildtype codes for a monomer, the fusion protein can dimerize and autophosphorylate its kinase domain [67,68].

Interestingly FGFR-fusions seem to play a role in other tumors of the central nervous system, especially in the pediatric and young adult population. Ryall et al. screened 1000 pediatric low grade gliomas and found FGFR-TACC fusions in 6.1% of them, which all had the diagnosis of pilocytic astrocytoma [69]. When they screened posterior fossa tumors of glial origin for the prevalence of FGFR1 mutations, Sievers et al. found a subset of nine tumors, in which 75% had a FGFR1-TACC1 fusion and could be distinguished in their gene expression pattern from other molecular phenotypes, so that the group even suggested defining these as a new molecular subtype, called “pediatric-type oligodendroglioma” [70]. There are several reports of FGFR1-TACC1 fusion in spinal lower grade glioma. Perwein et al. reported FGFR1-TACC1 fusion in 11.5% (3 out of 26) of their dataset of pediatric spinal lower grade glioma [71]. Additionally there are two case reports of young adults (22 and 32 years of age) with recurrent spinal lower grade gliomas: In the first one, a FGFR1-TACC1 was present at initial diagnosis [72], while the second patient developed FGFR1-TACC1 fusion only at recurrence of the tumor 14 years after the initial treatment by gross total resection [73].

In extraventricular neurocytoma, a 60% prevalence of FGFR1-TACC1 fusion is reported, and FGFR2-CTNNA3 fusion can typically be found in PLNTY [50,74]. Linzey et al. report a case of a 10-year-old patient with thalamic oligodendroglioma harboring FGFR3-PHGDH fusion who responded to treatment with the tyrosine kinase inhibitor ponatinib [75]. All these reports suggest an important role of FGFR-fusions as oncogenic drivers in several rare CNS tumors. However, data is sparse, and further research should be performed.

## 5. Expression and Detection of FGFR-Fusions in Gliomas

In 2012, Singh et al. screened nine primary glioma cell cultures for the prevalence of fusion genes by RNA sequencing and found six genetic rearrangements, of which five could be validated by PCR [10]. Afterwards they screened the dataset of The Cancer Genome Atlas (TCGA) and found specimens of 88 patients with primary glioblastoma and looked for the prevalence of these five fusion genes. The analysis revealed that the only recurrent fusion gene, which could not only be detected in cell cultures, but also in tissue specimens, was the fusion of FGFR3-TACC3, with the breaking points at exon 17 in the FGFR3 and at intron 8 of the TACC3 gene (5′-FGFR3ex16-8exTACC3-3′). In this dataset, an FGFR3-TACC fusion was found in two patients, and a concurrent fusion of FGFR1-TACC1 was seen in another sample. In summary, 3.1% of all tested glioblastoma samples harbored FGFR-TACC fusions [10]. Following this report, other groups screened their datasets for FGFR-TACC fusion genes: Di Stefano et al. found 17 of 584 glioblastomas (2.9%) and 3 out of 221 lower grade gliomas (3.5%) to be positive for FGFR3-TACC3 fusion. Additionally, they found one FGFR1-TACC1 fusion in a lower grade glioma [76]. Interestingly, they also had the opportunity to examine a tumor specimen of a recurrent glioblastoma, which had tested FGFR3-TACC3 positive at the time of initial diagnosis. The patient had received concomitant radiochemotherapy with temozolomide, according to the STUPP-regimen [5], before tumor progression. Biopsy of the recurrent tumor revealed the FGFR3-TACC3 gene fusion unaltered after therapy [76]. Among the FGFR3-TACC3 fusion genes, a high variability of breaking points, resulting in several FGFR3-TACC3 isoforms, was observed [31,76]. However, all fusion genes shared the intact kinase domain of FGFR3 and the coiled coil domain of TACC3.

Several studies were performed to detect FGFR-TACC fusions in human glioma and glioblastoma datasets. The prevalence of FGFR3-TACC3 fusions in human glioblastoma ranges from 1.3% [77] to 11.8% [78]. Mata et al. reported that 4.1% of 906 investigated glioblastoma samples harbored FGFR3-TACC3-fusions. They also performed a meta-analysis for the reported prevalence of FGFR3-TACC3 fusions at that time, stating an overall estimated prevalence of 3.7% for FGFR3-TACCT-fusions in glioblastoma [60]. We summarized the literature and the report the prevalence of FGFR3-TACCs fusions in Table 1.

With the increasing number of studies, more and more isoforms of the FGFR3-TACC3 fusion gene were observed. Today, at least 14 different genetic rearrangement of the gene are known [81]. The most common FGFR3-TACC3 fusions are the FGFR3ex17-TACCex11 form, followed by FGFR3ex17-TACC3ex10 and FGFR3ex17-TACC3ex8, which made up 79% of the fusion variants in the population studied by Di Stefano et al. [81]. Despite the variability in their length, ranging from 1706 bp (FGFR3ex18-TACC3ex4) to 805 bp (FGFR3ex18-TACC3ex13), they all shared an intact kinase and coiled coil domain [81]. All FGFR3-TACC3 fusions expressed in human gliomas consist of the FGFR3IgIIIc isoform [20,31].

Whether the fusion variants also differ in their cell signaling function is not fully understood, but especially after the observation of Nelson et al. that the isoforms do not necessarily share the same Y-residues for phosphorylation and might therefore play variable roles in signal transduction, this could be from immense interest [67].

Due to the high variability of FGFR3-TACC3 isoforms, pathological diagnosis of the gene rearrangement is particularly difficult: Most fusion genes, such as the EML4-ALK or BCL-ABL1 fusion in other cancer types, can be detected by FISH. Fusion genes, which lead to the overexpression of a specific gene, can be diagnosed by microarray [20,82]. Due to the natural proximity of the FGFR3 and the TACC3 wildtype gene, FISH is not an option for detection of the gene fusion, and due to the variety of expressed gene products, neither is microarray technology [20]. Kurobe et al. were able to develop an RNA-FISH assay to detect FGFR3-TACC3 fusion in bladder cancer; however, this is currently not used in clinical practice [83].

Most groups detected FGFR-TACC fusions by reverse-transcriptase PCR and afterwards, validation of the specific sequence of the fusion genes via Sanger sequencing [31,38,58,76,80]. Comparing the two methods in 101 preselected glioblastomas, Schittenhelm et al. showed that RT-PCR only had a sensitivity of 67% and a specificity of 100%, compared to next generation sequencing [80].

Early after the discovery of FGFR3-TACC3 fusions in human glioma, Di Stefano et al. observed that fusion-positive glioblastomas were highly positive in immunostaining with antibodies targeting the N-terminus of FGFR3 or the C-terminus of TACC3 [76]. This observation was confirmed by several other authors and is explained by reduced negative regulation of the fusion gene, as it lacks a specific sequence in the 3′-untranslated region of FGFR3 and is thereby able to escape miRNA mediated downregulation [31].

Bielle et al. reported a specificity of FGFR3-immunostaining as a predictive tool for the detection of the FGFR3 fusion of 92% with a sensitivity of 100%, leading to a positive predictive value of 56% and a negative predictive value of 100% [84]. Granberg et al. performed immunostaining for FGFR3 in 676 gliomas, of which 85% were completely unstained, 10% showed weak, 3.1% showed moderate, and 1.8% showed strong staining for FGFR3 [85]. With targeted sequencing for FGFR3 fusions in 51 intensely stained glioblastoma specimens, 41 harbored FGFR3 fusion gene rearrangements, leading to a sensitivity of 100% and a specificity of 88% for FGFR3-immunostaining [85]. Schittenhelm et al. determined a sensitivity of 95% and a specificity of 100% of FGFR3-immunohistochemistry for FGFR3 fusion detection (positive predictive value 67% and negative predictive value 100% in *n* = 88 specimens) [80].

Given that next generation sequencing requires a lot of effort and is relatively expensive, which relegates the diagnosis of FGFR3-TACC3 fusions to larger facilities that can provide the hardware and financial resources for the technical process, immunostaining might serve as a broadly available and affordable screening tool.

## 6. Histomolecular and Clinical Characteristics of FGFR3-TACC3 Fused Glioblastoma

There are several recurrent molecular and histological characteristics that were observed in an increasing number of investigations around FGFR3-TACC3 fusion-positive glioma: even in studies that were not limited to IDH-wildtype or former primary glioblastomas WHO grade 4, but included all forms of malignant gliomas, all gliomas that were positive for a FGFR3-TACC fusion presented the wildtype version of the IDH1 and IDH2 gene [10,20,31,58,60,81,84]. No FGFR3-TACC3 fusions were reported in oligodendroglioma, suggesting a restriction of the fusion to the astrocytic lineage of human glioma [58].

FGFR3-TACC3 fusions were mutually exclusive with EGFR-amplifications, a mutation that is present in about 80% of IDH-wildtype glioblastoma [58,60,76,81]. Mata et al. was the only group that found an EGFR-amplification in a FGFR3-TACC3 fusion-positive glioma [60]. Di Stefano et al. reported a trend towards the absence of the EGFRvIII variant that goes hand in hand with higher signaling activity of the EGFR-tyrosine kinase in FGFR3-TACC3 fusion-positive gliomas [81]. Furthermore, other authors reported an exclusivity of FGFR3-TACC3 fusions with amplifications of other tyrosine kinase receptors such as PDGFR, KIT, and MET [60,84,85].

FGFR3-TACC3 fusions were mutually exclusive with ATRX-loss and H3F3A mutations [60,80]. They occurred with less probability of harboring a co-occurring oncogenic TP53 mutation [60,80]. Regarding the frequency of CDKN2A inactivation, TERT-mutation, and cell cycle associated pathways, no difference in FGFR3-fusion-positive gliomas was described [60]. They were associated with a higher expression of stemness markers such as OLIG2 and GFAP [84].

While MDM4 alterations were absent in FGFR3-TACC3 fusion-positive gliomas, there is an association with a higher prevalence of MDM2 alterations and CDK4 amplifications, which can be found in 19% and 10%, respectively, of fusion-positive glioblastoma [84]. Even though CDK4 and MDM2 amplification itself is a favorable prognostic factor for the survival of glioblastoma patients, a subgroup of FGFR3-fusion-positive patients with CDK4 and MDM2 amplifications had an even better overall survival rate than those with CDK4 and MDM2 amplification that were negative for FGFR3 fusions [81].

Mata et al. performed methylation epic arrays on FGFR3-TACC3 positive glioblastoma, and their methylation profile most likely corresponded to the RTKII or mesenchymal subclass phenotype. In addition, FGFR3-fusion-positive glioblastomas harbored a generally lower overall mutational burden [60].

Remarkably, FGFR3-TACC3 positive gliomas were reported to possess specific morphological features that might reflect an initial step of tumorigenesis, as they can not only be found in glioblastoma, but also in FGFR3-fusion-positive lower grade glioma [84]. The specific recurrent morphological features of these tumors include monomorphous ovoid nuclei with nuclear palisading and attachment of the tumor cells towards blood vessels by parallel thin cytoplasmic processes, forming vague pseudorosettes [84]. Isolated tumor cells present with ovoid nuclei and ovoid cytoplasm and infiltrate the neuropil. A network of small capillary like vessels, arranged in an endocrinoid network, and spindled neoplastic cells embedded in a loose, myxoid background, with a tissue culture like appearance, is described as a “chickenwire pattern” by Broggi et al. [86]. In glioblastoma, these specific features were associated with different areas of higher cellular density, with anisocytosis, microvascular proliferation, and necrosis [84]. Furthermore, the tumors presented lower mitotic activity and signs of desmoplasia, such as CD34 labelling and microcalcifications, making differential diagnosis between astroblastoma, ependymoma, and angiocentric glioma, which share these morphological features, challenging [84]. These specific morphological features are reported to be present, at least focally, in 73% of FGFR3-fusion-positive tumors; however, as gliomas are known to be very heterogeneous, not all tumor areas reflect these features [84]. Gliani et al. investigated six FGFR3-TACC3 fusion-positive glioblastomas and five of them shared the described morphological characteristics [87]. Based on this observation, the group conducted prospective testing for FGFR3-TACC3 fusions in gliomas presenting with typical histomorphology. Two of the investigated tumors turned out to be negative for an FGFR3 fusion (the number of investigated cases is not published), concluding that the described recurrent morphological features are often shared by FGFR3-fusion-positive glioma, but the specificity for the molecular subtype is limited [87].

The epidemiology of FGFR3-TACC3 fusion-positive glioblastoma is mostly unspecific. However, Granberg et al. reported a female predominance, and Bielle et al. reported a sex ratio of 1:1 for this specific type of glioblastoma [84,85], which is interesting, as glioblastoma are usually more frequent in men [11]. The presence of a FGFR3 fusion in glioblastoma IDH wildtype is reported as a favorable prognostic factor, associated with a better overall survival compared to glioblastoma IDH-wildtype without FGFR3 fusion [88]. However, the survival rate is not better than the survival rate for IDH-mutated gliomas [81]. The analysis of imaging data of a large set of FGFR3 fusion-positive glioblastoma revealed an association with the occurrence in the cortical-subcortical region, insular, and temporal lobe location, which might be due to the specific role of FGFRs in the development of these brain areas [81]. FGFR3 fusion-positive tumors presented with recurrent radiogenomic features, including a less frequent eloquent location, poorly defined contrast-enhancing and non-enhancing tumor margins, as well as increased edema in glioblastoma and poorly defined tumor borders in lower grade glioma [81,89] Radiomic analysis for the classification of the FGFR3-TACC3 status of glioblastoma led to an area under the curve (AUC) of 0.87 in the first dataset and 0.754 in a second validation set, allowing the conclusion that FGFR3-TACC3 positive glioblastoma has a distinct radiomic signature [81].

## 7. The Functional Role of FGFR3-TACC3 in Glioblastoma Cells

The FGFR3-TACC3 fusion protein is involved in several cellular processes and signaling cascades, leading to FGFR3-TACC3 overexpression, increased kinase activity, and corresponding downstream signaling, morphological changes, increased cell growth, altered cellular metabolism, stress response, and even dysregulated mitotic progression, resulting in aneuploidy.

In the very first study on FGFR3-TACC3 fusion, Singh et al. transfected fibroblasts and astrocytes with the FGFR3-TACC3 fusion gene and observed not only anchorage independent growth of these cells in soft agar, but also a gain of proliferative capacity and the formation of glioma lesions, expressing the glioma stem cell markers OLIG2, phosphohistone H3, nestin, and GFAP, while EGFRvIII transfected astrocytes did not show these markers [10]. Stimulation with FGF did not affect the downstream signaling of FGFR3-fusion-positive cells, but instead a constitutive phosphorylation of the tyrosine kinase domain of FGFR3 and FRS2 was observed [10]. All of these effects were abolished by the tyrosine kinase inhibitor PD173074.

The kinase activity of the FGFR3-TACC3 fusion protein plays a crucial role in the proliferation and survival of glioblastoma cells. The ability to dimerize TACC3 leads to the presence of a constitutively dimerized, and thereby activated, FGFR3 tyrosine kinase domain in fusion-positive tumor cells and hyperactivated downstream signaling, resulting in an overexpression of phosphorylated FRS2, the initial intracellular binding partner of FGFR3, to activate ERK1/2 and AKT signaling [20,22,27,40,68]. Parker et al. detected an increased activation of pERK, but not of STAT3 and pAKT, in fusion-positive cells while other authors saw enhanced STAT3 and STAT1 activation [31,40]. Nelson et al. showed that only plasma membrane localized FGFR3-TACC3 fusion protein leads to formation of oncogenic foci in fusion transfected NIH3T3 cells. This goes along with increased MAPK signaling activation, while cytoplasmatic localized FGFR3-TACC3 does not induce oncogenic transformation, supporting the hypothesis that the oncogenic force of the fusion protein is dependent on its kinase activity [68]. The phosphorylation site Y746 is of major importance for activation of ERK, STAT and PI3K signaling. Y746 is hyperphosphorylated in FGFR3-TACC3 fusion-positive cells and MAPK activation is increased in fusion-positive cells [67]. Treatment with the kinase inhibitors BGJ398 and trametinib resulted in reduction in MAPK signaling and had an antitumor effect in FGFR3-TACCex11 and FGFR3-TACC3ex8 transfected cells, but the signaling and treatment response differed between the two isoforms [68].

TACC3 in its wildtype from is phosphorylated by Aurora-A and forms a complex with clathrin and ch-TOG, which is localized to the mitotic spindle apparatus and provides its stability [68]. When fused to the FGFR3-protein, TACC3 lacks a phosphorylation site for Aurora-A [68]. Confocal imaging of FGFR3-TACC3 showed an arc-shaped structure of the protein, bending over and enchasing the metaphase spindle poles, but not relocating to the mid body, leading to erratic mitotic segregation [10]. FGFR3-TACC3-positive cells exhibit a 3 to 5-fold higher number of errors in chromosomal segregation and resulting aneuploidy, an effect, that could be reduced to 80% by treatment with the kinase inhibitor PD173074 [10]. This, however, implies that the activity of FGFR3-TACC3 in aneuploidy induction is dependent on its kinase activity.

Gene ontology mapping of the dataset from TCGA showed enriched expression of genes that are related to oxidative phosphorylation, high mitochondrial activity, and biogenesis [90]. Additionally, increased mitochondrial DNA, mitochondrial mass, and higher levels of ATP were detected in FGFR3-TACC3 transfected astrocytes compared to control cells, and the fusion-positive cells showed an elevated basal and maximal oxygen consumption rate, as well as a mild increase in the extracellular acidification rate [90]. Gene expression levels of the respiratory complex proteins SDHB, UQCRC1, ATP5A1, and the mitochondrial membrane transporter VDAC1 were elevated in fusion-positive cells [90]. Anti-pY immunoprecipitation showed that only FGFR3-TACC3 fusion-positive cells contained phosphorylated PIN4, an activator of mitochondrial metabolism and anabolic response, leading to accumulation of reactive oxygen species and thereby, elevated expression of the transcription regulators PGC1α and ERRγ, which increase mitochondrial metabolism [90]. Acordingly, the FGFR3-TACC3-positive cells were sensitive towards treatment with mitochondrial inhibitors such as menadione, metformin, and tigecycline in vitro [90].

Besides this, FGFR3-TACC3 fusion leads to morphological changes characterized by the rounding up of the cells in HEK293T cells, the activation of cell signaling pathways related to chaperone activation, the stress response and regulation of tp53 expression, and the degradation and resistance to EGFR inhibitors in HNSCC and urothelial carcinoma cells [40,91,92].

## 8. FGFR3-TACC3-Fusions: A New Hope for Targeted Therapy?

Several in vitro and in vivo studies regarding treatment with FGFR-Inhibitors for FGFR3-TACC3 fusion-positive gliomas have led to promising results. The tyrosine kinase inhibitors AZD4547, PD173074, BGJ398, and JNJ-42756493 inhibited the proliferation of FGFR3-TACC3 transfected astrocytes in vitro and led to reduced tumor growth and prolonged survival in glioma-bearing mice [10,11,39,76]. PD173074 was even more able to suppress the kinase phosphorylation of FGFR3 and reduced aneuploidy by about 80% [10]. In FGFR3-TACC3 fusion-positive glioma cell lines, the FGFR-inhibitors PD173074 and AZD4547 had an antiproliferative effect, and the kinase inhibitor pazopanib caused cell cycle arrest [40]. Furthermore, a higher sensitivity of fusion-positive glioma cells towards the MEK/ERK inhibitor U0126 was observed [20].

In other cancer cell lines, such as cholangiocarcinoma, nasopharyngeal carcinoma, and urothelial carcinoma, a higher potency of tyrosine kinase inhibitors has also been described [54,93,94,95].

Parker-Kerrigan et al. transfected 10 unique siRNAs into glioblastoma and bladder cancer cell lines, which led to the depletion of the FGFR3-TACC3 fusion protein, while wildtype FGFR3 was not affected [96]. The siRNA knockdown of FGFR3-TACC3 in glioblastoma cells leads to reduced cell growth in vitro and in vivo [40,96].

A small number of case reports regarding the use of kinase inhibitors in patients with FGFR3-TACC3 fusion-positive glioblastomas exists: Wang et al. treated two patients with the kinase inhibitor anlotinib [97]. A 65-year-old woman with a FGFR3-TACC3 fusion-positive glioblastoma, diagnosed in 01/2020, received anlotinib after she had tumor progress following gross total resection and treatment with radiochemotherapy with temozolomide and five adjuvant cycles of temozolomide. She had partial response after three months of treatment with anlotinib and was still alive at the time of publication [97]. The second patient, a 44-year-old woman, was diagnosed with FGFR3-TACC3-positive glioblastoma in 12/2017. She was treated with radiochemotherapy, according to the STUPP-regimen, and afterwards received treatment with nedaplatin and bevacizumab. In 06/2018, she had tumor progress and started treatment with a temozolomide rescue scheme and anlotinib. After two months, she had partial response and was also still alive when the authors published the article [97].

The kinase inhibitor JNJ-42756493 was also administered to two glioblastoma patients, a 52-year-old man and a 64-year-old woman, who were both diagnosed with FGFR3-TACC3-positive glioblastoma. Both patients underwent gross total resection of the tumor, followed by concomitant radiochemotherapy with temozolomide. After they experienced tumor progress, both received JNJ-42756493. The first patient reached stable disease for 115 days before he exhibited new tumor progression, while the second patient had partial response after 4 weeks and was on treatment for 134 days before she exhibited progressive disease [39]. The pan-FGFR inhibitor erdafitinib (JNJ-42756493) was administered to 65 patients with solid tumors in a phase 1 trial [98]. FGFR3-TACC3 fusions were detected in one glioblastoma patient, two urothelial carcinoma patients, and one patient with adrenal carcinoma who showed partial response under treatment with erdafitinib [98].

Currently, there are diverse ongoing trials regarding the use of tyrosine kinase receptors in FGFR3-TACC3 positive glioblastomas (Table 2).

The FGFR1-3 receptor inhibitors AZ4547 and BGJ398 were both evaluated for patients with FGFR1-TACC1 and FGFR3-TACC3 fused glioblastomas (NCT028224133 and NCT01975701); however, the results have not yet been published. A study considering infigratinib (BGJ398) for patients with unselected FGFR-altered glioblastoma proved the safety of the drug, despite only limited effects [99]. Nevertheless, response with a stable disease for over one year was observed in a patient with a FGFR3-TACC3-fused glioblastoma [99].

## 9. Conclusions

Currently, more than 50 new cancer drugs make their way through permission processes each year, and the description of more and more new biomarkers is leading to a completely new concept of histomolecular diagnostics and finally, therapeutic practice. While this offers new opportunities, this development also points out the necessity of proving the relevance and feasibility of diagnostic markers and possible therapeutic targets. The FGFR3-TACC3-fusion with a prevalence of 3–4% in human glioblastoma is relatively rare, especially in the background of the low incidence of glioblastoma in general. Nevertheless, committing to the concept of personalized medicine means that the characterization of small subgroups and their potential relevance for therapeutic approaches deserves research efforts.

FGFR3-TACC3 fusion-positive glioblastoma are characterized by recurrent histological features, and they present with a distinct molecular profile. Their cellular signaling and also clinical behavior differs from that of IDH-wild-type glioblastoma of other tyrosine kinase receptors, and the existing data suggest a specific ontogenetic origin during tumorigenesis in the astrocytic lineage.

The FGFR3-TACC3 fusion gene has been shown to serve as a prognostic factor, so it would be reasonable to invest in the diagnostic effort in clinical practice. Although the final detection of the fusion gene via RNA-sequencing is expensive, screening via immunostaining is affordable and is an easily approachable method that could be used in less privileged facilities.

Data from in vitro and in vivo experiments, as well as case studies, suggest a better response of FGFR3-TACC3 fusion glioblastomas to treatment with tyrosine kinase inhibitors. Whether this is due to the specific biological activity of the fusion protein, or is due to the generally more favorable prognosis of fusion-positive glioblastoma patients, needs to be evaluated. The low frequency of the genetic alteration makes recruitment of adequate patient cohorts difficult, leaving umbrella studies as the only practicable rationale.

In conclusion, FGFR3-TACC3 fusion might be an emerging opportunity for personalized diagnostic and targeted therapy in glioblastoma. Its specific functional and clinical relevance should justify further effort to answer open questions and establish the markers in daily clinical practice.

## Figures and Tables

**Figure 1 ijms-23-08675-f001:**
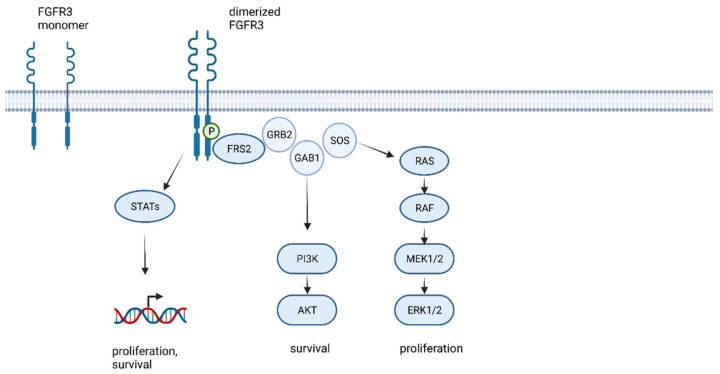
FGFR kinase signaling and activation of MAPK pathways leading to cell proliferation and cell survival (created with biorender.com, accessed on 21 May 2022).

**Figure 2 ijms-23-08675-f002:**
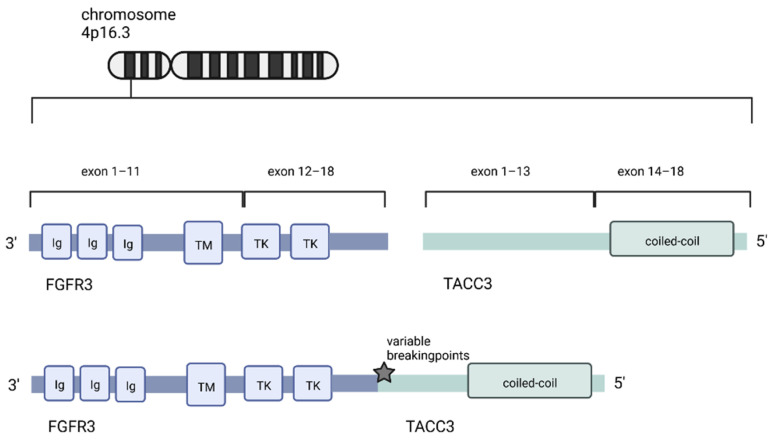
The FGFR3 and the TACC3 gene are both located on chromosome 4p16.3. The oncogenic fusion gene codes for FGFR3, harboring an intact kinase domain fused to the TACC3 gene, including the functioning coiled-coil domain (created with biorender.com, accessed on 21 May 2022).

**Table 1 ijms-23-08675-t001:** Reported prevalence of FGFR3-TACC3 in human glioma.

Original Article	Prevalence of FGFR3-TACC3 Fusion	Reference
Singh at al.	2.1%	[10]
Di Stefano et al.	2.9%	[76]
Parker et al.	8.3%	[31]
Bao et al.	5.1%	[79]
Asif et al.	11.8%	[78]
Na et al.	1.3%	[77]
Mata et al.	4.1%	[60]
Yoshihara et al.	4.4%	[59]
Ferguson et al.	2.6%	[58]
Schnitthelm et al.	2.3%	[80]
Di Stefano et al.	2.5%	[81]

**Table 2 ijms-23-08675-t002:** Currently ongoing clinical trials for treatment targeting FGFR-signaling in human glioma. Not all trials are limited to FGFR-fusion-positive tumors (www.clinicaltrials.gov, accessed on 13 May 2022).

NCT Number	Drug	Conditions	Status
NCT05222165	Infigratinib	advanced solid tumors, CNS tumors, or progressive LGG with selected FGFR1-3 alterations	not yet recruiting
NCT05267106	Pemigatinib	previously treated GBM or other primary CNS tumors with FGFR1-3 alterations	recruiting
NCT04945148	Metformin	GBM, IDH-wild-type	not yet recruiting
NCT04424966	Infigratinib	recurrent glioma	recruiting
NCT03210714	Erdatifinib	advanced solid tumors	suspended
NCT02465060	various TKI	advanced solid tumors	recruiting
NCT04004975	Anlotinib	recurrent GBM	unknown status
NCT03155620	various TKI	advanced solid tumors	recruiting
NCT04547855	Anlotinib	GBM	recruiting
NCT05033587	Anlotinib	MGMT-unmethylated GBM	recruiting
NCT04216550	Apatinib	recurrent glioma	recruiting

## Data Availability

Not applicable.

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
