# Peer review of "FGFR3-TACCs3 Fusions and Their Clinical Relevance in Human Glioblastoma"

_ijms, 2022, doi:10.3390/ijms23158675_

Round 1

Reviewer 1 Report

The manuscript by Gott and Url is a comprehensive review about FGFR3-TACCs fusion in cancer. The review is scientifically sound and well written. I just have two points that should be clarified/improved prior publication of the manuscript:

1. The authors should cite original papers demonstrating certain findings whenever possible and not the reviews, for example lines 76 and 77 the authors refer to review about FGFR interaction with cadherins and galectins and not original publications. This should be improved for the whole manuscript (especially since it is a review that should refer to original work).

2. Is there anything known about fusions of other FGFRs in cancer? This point should be discussed as well.

Reviewer 2 Report

The work of Gott et al. clarified FGFR3-TACC3 fusions and their clinical relevance in human glioblastoma. Although the review is well written, I have some suggestions before acceptance.

Comments

- Add a comprehensive paragraph about glioblastoma after Introduction

- The authors considered the studies of 2022? This is an important point in order to provide a review as updated as possible

Round 2

Reviewer 2 Report

Nothing to add. For me the manuscript is now acceptable.